# UAV Path Planning in Multi-Task Environments with Risks through Natural Language Understanding

Chang Wang [1], Zhiwei Zhong [2,*], Xiaojia Xiang [1], Yi Zhu [3], Lizhen Wu [1], Dong Yin [1] and Jie Li [1]

[1] College of Intelligence Science and Technology, National University of Defense Technology, Changsha 410073, China
[2] College of Computer Science and Electronic Engineering, Hunan University, Changsha 410082, China
[3] School of Information Engineering, Nanjing Audit University, Nanjing 211815, China
[*] Correspondence: zhongzhiwei1987@hnu.edu.cn

**Abstract:** Path planning using handcrafted waypoints is inefficient for a multi-task UAV operating in dynamic environments with potential risks such as bad weather, obstacles, or forbidden zones, among others. In this paper, we propose an automatic path planning method through natural language that instructs the UAV with compound commands about the tasks and the corresponding regions in a given map. First, we analyze the characteristics of the tasks and we model each task with a parameterized zone. Then, we use deep neural networks to segment the natural language commands into a sequence of labeled words, from which the semantics are extracted to select the waypoints and trajectory patterns accordingly. Finally, paths between the waypoints are generated using rapidly exploring random trees (RRT) or Dubins curves based on the task requirements. We demonstrate the effectiveness of the proposed method using a simulated quadrotor UAV that follows sequential commands in four typical tasks with potential risks.

**Keywords:** UAV; natural language understanding; path planning; RRT; Dubins curve

## 1. Introduction

Path planning is the fundamental capability for an autonomous UAV to carry out real-world tasks such as logistics [1], warehouse construction [2], surveying [3] and environmental monitoring [4], agriculture [5] and localization [6], and so on. Various methods have been proposed for solving the UAV path-planning problem, e.g., genetic algorithm (GA) [2], particle swarm optimization (PSO) [7], deep reinforcement learning (DRL) [8], A* and rapidly-exploring random Tree (RRT) [9], et al. For example, local optima were avoided by introducing nonlinear dynamic inertia weights into traditional PSO [7]. Considering environmental risks such as enemy radar detection and missile attack, the dueling double deep Q-networks (D3QN) algorithm was designed for action selection based on a situation assessment model [8]. By comparing two typical graph-based and sampling-based algorithms, i.e., A* and RRT, their limitations were discussed, and they were improved by ripple reduction and smoothing, respectively [9]. Recently, visual perception has been integrated in the control loop of the UAVs, and deep neural networks have been used to learn various sensorimotor skills in real-world environments, such as high-speed indoor racing [10], acrobatics (the power Loop, the Barrel Roll, and the Matty Flip) [11], swarming in a forest [12], among others. Due to the limitation of onboard computation resources, trajectories have been optimized by leveraging relative gate observations [10], demonstrations from an optimal controller [11], or geometrical configuration constraints [12].

Although artificial intelligence (AI) technologies have been developed to achieve higher UAV autonomy, human supervision is still necessary to guarantee task efficiency as well as UAV safety, as discussed in our previous work [13]. Specifically, the human operator of the ground control station (GCS) needs to handle dynamic events (e.g., new task locations or emerging risks) in multi-step tasks such as reconnaissance and surveillance. The human

workload would dramatically increase with the number of UAVs as well as task complexity. Compared with traditional human-machine interfaces (e.g., keyboard, mouse, and touch screen) [14], voice and gestures are more effective for controlling multiple UAVs [15]. As for UAV path planning, augmented reality (AR) [16] and human action recognition [17] can also be used. However, natural language is still considered to be the most convenient way for an end-user to control an intelligent vehicle in urgent situations [18]. Based on our previous work [15], we will further investigate how to automatically generate UAV paths based on human voice commands. We note that we focus more on the interactive perspective of UAV path planning rather than developing cutting-edge sensorimotor control or path-planning algorithms as done in [10–12].

Natural language processing (NLP) includes several topics such as automatic speech recognition (ASR) [19], natural language understanding (NLU) [20], and machine translation [21], et al. Current state-of-the-art ASR systems are typically end-to-end based on autoregressive models [22] or deep learning models [23]. Voice command recognition can be used for controlling robots in simple robotic tasks using hidden Markov models (HMMs) [24] or deep learning [25]. For example, the answer-set rules were designed, and the commands were converted into a sequence of actions for robot task planning [26]. In another work, a UAV was controlled by a domain-based speech-to-action method [27]. On the other hand, NLU is more applicable for multi-task scenarios that involve information gathering, question answering and dialogue management. It is essential to extract the semantic elements from human utterance [28]. For example, human instructions were inferenced by a probabilistic generative model for path planning [29,30]. The number of command words and the difficulty of speech recognition were both reduced by using location names instead of coordinates. However, assumptions were made that the accuracy of speech recognition was sufficiently high.

In this paper, the main contributions are as follows:

(1)　We propose a novel interactive framework for automatic path planning with a multi-task UAV through the understanding of compound natural language commands.
(2)　We propose a multi-task command understanding method using RNN-based tagging and semantic annotation, which can extract keywords that describe the task types and the task requirements instructed by the human operator.
(3)　We propose a novel algorithm to efficiently select the start and the exit waypoints for each task zone from a small set of candidate waypoints according to the tasks.

The rest of the paper is organized as follows. Section 2 analyzes the problem. Section 3 explains the proposed method. Section 4 discusses the simulations and results. Finally, Section 5 concludes the paper.

## 2. Problem Statement

The choice of path-planning algorithms largely depends on task requirements and environmental characteristics. For example, RRT-based or A*-based algorithms are possible choices in simple navigation tasks [9]. In a coverage search task without any risks, the optimal pattern of a UAV's scanning lines can be planned along with its flying height and the angle of its camera constrained by its field of view (FOV). In cluttered urban environments, a risky-aware planning strategy can be developed to speed up the task while minimizing the risk cost [31]. In a reconnaissance task with potential risks such as radars or ground-to-air missiles [8,13], deep reinforcement learning algorithms can be used to find the optimal path that maximizes the UAV's accumulated rewards.

However, it has been assumed that a UAV is fully aware of the situation, and it can autonomously adapt to environmental changes or task dynamics in a responsive manner. This is not realistic in real-world applications that unexpected events can occur, which is beyond the understanding of machines. In such cases, human intervention is necessary to guarantee UAV safety while solving the given tasks. Typically, such events need to be handled as soon as possible, and it is inefficient to manually select new waypoints using a

mouse, a keyboard or a touch screen. Therefore, we propose to solve the path-planning problem using natural language commands that are more convenient and informative.

This is still challenging due to several reasons. First, human utterances can be redundant, with irrelevant information, and the speech recognition results can be nonsense with keywords or parameters missing. Therefore, we need to design a concise format of natural language commands that can convey necessary information as much as possible without ambiguity or mistakes. Second, the commands can be compound- that is, they involve several tasks or sequential actions. Each task or action may correspond to a trajectory pattern such as a straight line, irregular line, circle, or curve. As a result, several path-planning algorithms are needed, and the paths must be connected coherently with consideration of UAV kinematics. In other words, we need to design a flexible algorithm that can orchestrate several flyable paths in a unified manner. Finally, the symbol grounding problem must be considered to correlate the uttered words with the entities and actions in a simulated or a real-world environment. In our case, we need to model the task zones and define the actions to enable automatic UAV path planning.

## 3. Method

### 3.1. System Framework

The system framework of the proposed method is shown in Figure 1. First of all, the semantic extraction module is responsible for processing natural language commands. Keywords about the task types and the requirements have to be extracted and ordered in the form of a task sequence. Second, the task configuration module correlates the commands with the task zones in a given map, and a small set of candidate waypoints are generated for each task zone accordingly. We note that the constraints of waypoints satisfy the requirements of human intentions, e.g., flying around a tree or a radar should be different. In this way, path planning can be more efficient than searching in a large space. Finally, the path planning module can generate a UAV trajectory by selecting the waypoints and orchestrating them with lines or curves. We note that any path planning algorithms can be used. However, we choose specific algorithms in this paper for the reconnaissance and surveillance tasks with no-fly zones.

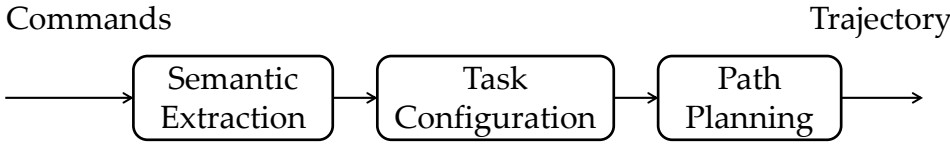

**Figure 1.** System framework of the proposed path planning method.

### 3.2. Task Zone Modeling

There are several kinds of task zones with risks for a multi-task UAV [32]. On one hand, high risks must be avoided to guarantee the UAV safety. For example, obstacle avoidance is a fundamental capability for a UAV to avoid colliding with any objects that may cause a deadly crash, e.g., trees, birds, or another UAV [33]. In addition, a UAV must avoid no-fly zones with radar or ground-to-air missiles, and it is better to keep away from these zones by following optimal policies as discussed in our previous work [8]. In the above cases, UAV path planning must be carried out in a responsive manner as fast as possible. On the other hand, low risks exist in the tasks without counterattacks except for unexpected dangers. For example, a quadrotor UAV searches over an open area to find wounded hill climbers, or hovers over a flock of sheep to keep track of their traces. In these cases, UAV path planning can be more persistent without many changes. Table 1 lists the abovementioned risks and proposes the corresponding approaches of modeling and solutions.

**Table 1.** Modeling and solutions for task zones.

| Task Type | Task Zone Modeling | Solution |
|---|---|---|
| Avoid obstacles | Cylinder, Cuboid | Bypass closely |
| Avoid radar or missile | Hemisphere | Bypass far enough |
| Reconnaissance | Rectangle, Circle | Coverage search |
| Surveillance | Circle | Hover tracking |

For obstacles such as trees or buildings, we use cylinders and cuboids to represent the envelop zones around them. We assume that the obstacles are static, and the parameters of their sizes are known to the UAV in a given map. In this case, the UAV is expected to bypass closely around the obstacles without the need of keeping too far away from them. In contrast, we use hemispheres to model the risky zones around the radars and missiles, and the UAV must bypass them far enough for safety reasons. We note that the radius of a hemisphere is also known. As a result, typical path planning algorithms can be used to avoid these risky zones.

For targets such as ground vehicles, humans or livestock, we use rectangles or circles to model the task zones around them. For example, scanning lines are typically used for coverage search of targets in a reconnaissance task. If a specific target has to be tracked in a surveillance task, a quadrotor UAV can hang in the air or a fixed-wing UAV can hover in a circle. The parameters of the scanning lines or circles are related to the visual sensor parameters of the UAV. In this paper, we also assume they are known for path planning.

Under the above assumptions and discussions, we need to propose a path planning algorithm that can generate lines and circles for these risky zones and orchestrate them in a coherent manner. Moreover, we need to extract relevant locations and parameters for automatic path planning from the uttered commands.

### 3.3. RNN-Based NLU for UAV Path Planning

In the uttered sentences, words are organized with non-linear structures, which can be transformed into tree-like graphs. The main task of NLU is to reveal the dependencies between the words and obtain the tree structures of the sentences. A variety of methods have been proposed for NLU, including random forests [34], attention-based deep neural networks [35], co-interactive Transformer [36], graph LSTM [37], parallel interactive network (PIN) [38], and recurrent neural network (RNN) [39]. In this paper, we choose RNN to analyze the commands because RNN has good performance for discovering semantics from sequential data. Furthermore, RNN can handle utterances with acoustically confusing words [40]. Figure 2 illustrates an example of confusion network for the ground-truth utterance "avoid obstacle two".

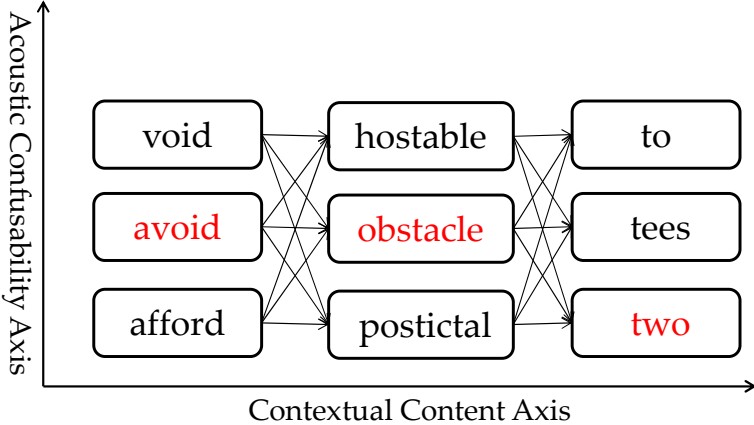

**Figure 2.** An example of confusion network for ground-truth utterance "avoid obstacle two".

The main component of RNN is a directed graph, i.e., digraph. The elements linked by chains in the digraph are called cyclic units. In general, the chained connections are comparable with the hidden layers in the feedforward neural network. The concept of layer in RNN refers to the cycle unit of a single time step. The learning data input in sequence are $C = C_1 + C_2 + \ldots + C_\tau$, where $\tau$ is the unfolded length of the RNN. At the time step $t$ the system status of the RNN can be represented as $h^{(t)} = f\left(s^{(t-1)}, C^{(t)}, \theta\right)$. In the view of dynamic system, the system status mainly describes the changes of all points in a given space with time steps, so it can be applied to the system state equation.

Moreover, $s$ is an internal status relevant with the system status $s = s(h, C, y)$. Since the solution of the current system state requires the use of the internal state data of the previous time step, the calculation of the cycle unit needs to be recursive. Under the influence of tree structure, all previous time step cycle cells are regarded as the parent node of current time step cycle cells. In the formula, $f$ refers to the excitation function or an encapsulated feedforward neural network. The former mainly corresponds to the simple cyclic network (SRN) structure, and the latter corresponds to the gating algorithm and commonly used deep learning algorithm. Hyperbolic tangent function and logistic function are the commonly used excitation functions.

In this paper, we design a structure of natural language commands in the sequential format of "action + location". As we only consider the path planning of one UAV in this paper, the subject "UAV" can be omitted. We will leave the discussion of multi-UAVs for future work. We annotate each compound command in the above format. Therefore, the resulting semantic sequence annotation can be checked if any necessary information is missing for path planning. If the human operator does not mention which entity in the environment should be avoided, searched or tracked, the previous action would be used for the current entity. If no previous actions were available, the human operator would be reminded to give the command again.

Sequence tagging is usually considered to be a fundamental problem for NLP. In other words, each word in a sentence needs to be tagged with a label in a linear manner. In the sequel, we illustrate how semantic tags are used to extract semantics based on an RNN model.

In reconnaissance and surveillance tasks, a typical command can be segmented into an annotated structure in two steps. First, each word is tagged using RNN. Specifically, a semantic tag set can be defined as {VE, AD, NO, CL}, where VE is a verb (e.g., bypass and search), AD is an adjective or adverb that describe how the action should be executed (e.g., static, fully, and slightly), NO is a quantifier (e.g., numbers), and CL is a noun (e.g., target and region). Second, relevant words are combined according to the predefined format of command annotation. In other words, a VE and an AD combine as a predefined action command, while an NO and a CL combine as a task location in the given map. For example, the command "bypass area 1, thorough search area 2, track target 3" can be annotated as shown in Figure 3.

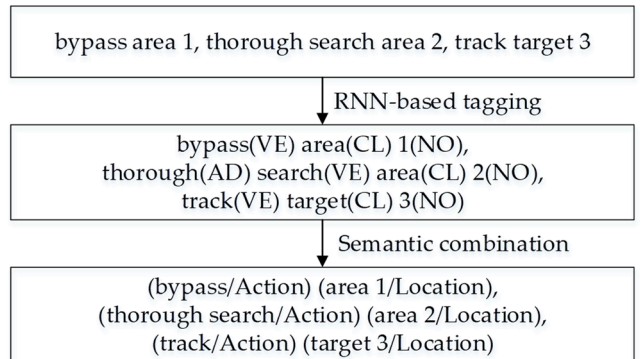

**Figure 3.** Command understanding through RNN-based tagging and semantic annotation.

Then, the annotations of actions and locations are grounded in a simulated or a real-world environment. For example, each action represents a defined trajectory such as a segmented line, circle, or combined. On the other hand, each location represents a defined coordinate in the given map. In this way, relevant information can be extracted from the annotated command for further path planning.

### 3.4. Path Planning with RRT and Dubins Curves

According to the analysis of risky zones and task solutions in Table 1, we choose Rapid-exploration Random Tree (RRT) and Dubins curves as the two candidates for UAV path planning. The reasons for this choice are as follows. On one hand, RRT is a sampling-based path planning method with high search efficiency in multi-dimensional spaces [9], it can balance the random exploration and the goal-directed exploitation in complex environments cluttered with obstacles. On the other hand, Dubins curves can handle the path planning problems with given start and end positions, along with corresponding moving directions that meet the requirements of the UAV kinematics, i.e., finding the shortest path with consideration of the UAV's turning radius [41]. We illustrate the ideas of RRT and Dubins curves in Figure 4.

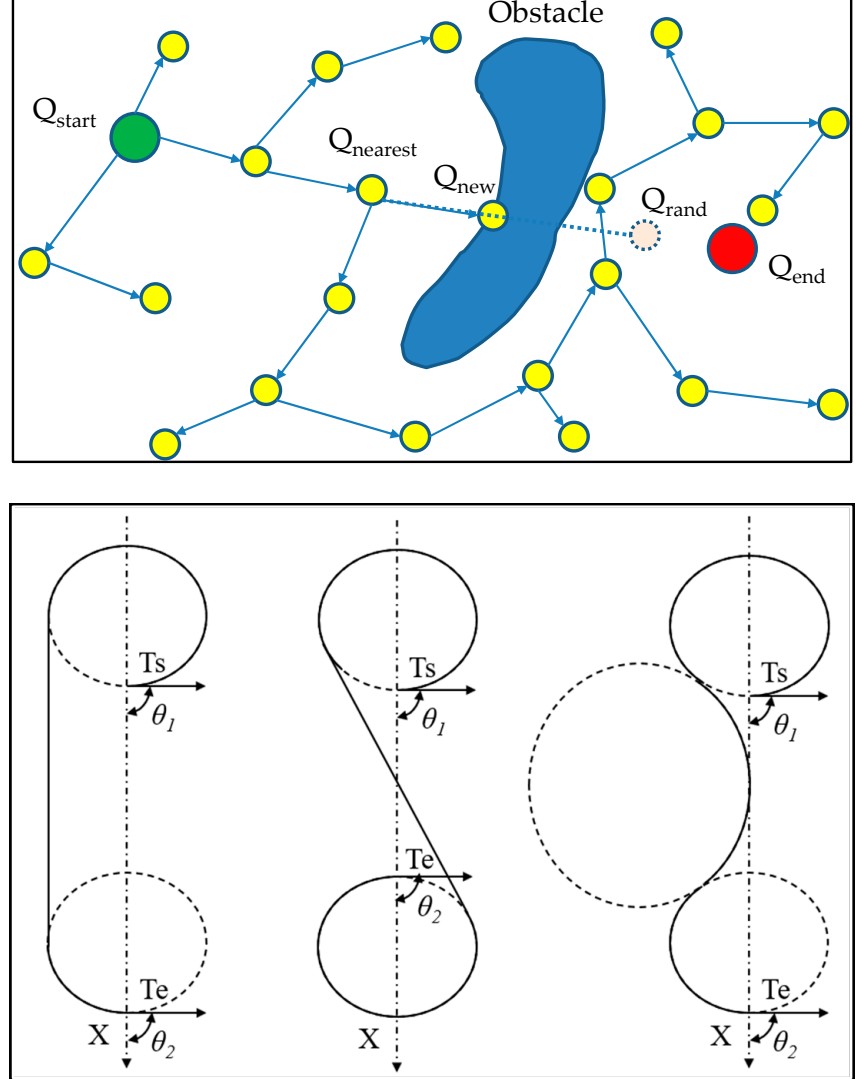

**Figure 4.** Illustration of RRT algorithm (**up**) and Dubins curves (**down**) that are LSL, LSR and LRL.

The main idea of RRT is to generate a tree-like path $T$ from the start position $Q_{start}$ to the end position $Q_{end}$. Before finding a new waypoint $Q_{new}$ at each step, a reference point $Q_{rand}$ is generated randomly from the nearest waypoint $Q_{nearest}$ in $T$ towards $Q_{end}$ or generated randomly in the free space unoccupied by obstacles. If $Q_{new}$ is accessible (i.e., not in the area of any obstacles), then it is added to $T$. Otherwise, a new $Q_{rand}$ would be generated. Finally, when $Q_{nearest}$ is close enough to $Q_{end}$, all the waypoints in $T$ are connected to obtain the planned path.

Dubins curves combine the maximum curvature arc (C) and the straight-line segment (S) to meet the UAV kinematic constraints [41]. For a given entry angle and an exit angle, a Dubins curve can either be CSC curve (LSL, LSR) or CCC curve (LRL), where L represents counterclockwise rotation to the left while R represents clockwise rotation to the right, see Figure 4.

In this paper, we consider UAV path planning with sequential tasks in risky environments as shown in Table 1. We note that RRT and Dubins curves are good choices for the four types of tasks (i.e., obstacle avoidance, radar or missile avoidance, reconnaissance, and surveillance) based on natural language understanding. On one hand, RRT is efficient to plan a path for any two waypoints with no obstacles in between or with convex shapes of obstacles (e.g., cylinders or cuboids). On the other hand, Dubins curves are effective for circular maneuvers required by the kinematics of UAVs, especially the fixed wings. The UAV path planning algorithm (WGS-NLU) is summarized as follows.

We assume that the start point ($S_0$) and the end point ($E_{n+1}$) are known, and the structured commands $\{Act_i, Loc_i\}$, $(i = 1, 2, \ldots, n)$ have been extracted through natural language understanding (NLU). Each command pair $\{Act_i, Loc_i\}$ corresponds with a task type. Denote by $\{Task_i\}$, $(i = 1, 2, \ldots, n)$ the corresponding sequence of tasks, and the risky zone $Z_i$ for $Task_i$ can be located in the grid map, see the dashed areas in Figure 4. Then, the set of candidate waypoints can be found around the task zone. For example, "$Act_i =$ bypass far enough, $Z_i =$ radar" means that the candidate waypoints $\{WP_k(Z_i)\}$ should be at least one grid away from $Z_i$ in all directions (the green solid circles), while "$Act_{i+1} =$ bypass closely, $Loc_{i+1} =$ obstacle" means the candidate waypoints $\{WP_k(Z_{i+1})\}$ can be the closest grid vertices to the obstacle in all directions (the gray solid circles).

As mentioned above, Dubins curves are chosen for avoiding radars (see the brown arrows connecting $S_i$ and $E_i$ in Figure 5), while RRT is chosen for avoiding obstacles (see the blue arrows connecting $S_{i+1}$ and $E_{i+1}$ in Figure 5). Then we use a distance criteria for selecting the start point $S_i \in \{WP_k(Z_i)\}$ and the end point $E_i \in \{WP_k(Z_i)\}$. If $i = 1$, then $S_i$ is the closest point in $\{WP_k(Z_i)\}$ to $S_0$; otherwise, $S_i$ is the closest point in $\{WP_k(Z_i)\}$ to the previous waypoint $E_{i-1}$. If $i = n$, then $E_i$ is the closest point in $\{WP_k(Z_i)\}$ to $E_{n+1}$; otherwise, $E_i$ is the closest point in $\{WP_k(Z_i)\}$ to the center $O_{i+1}$ of the next task area $Z_{i+1}$.

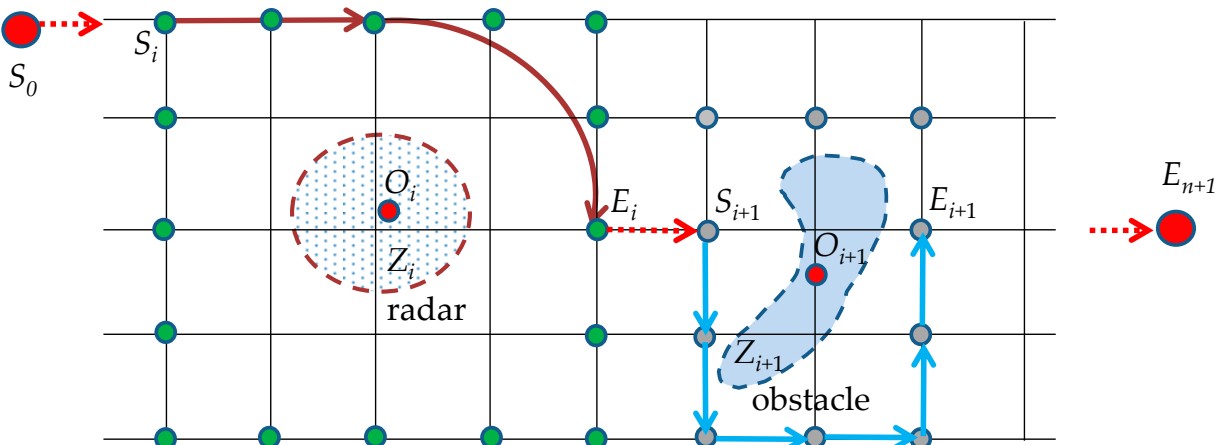

**Figure 5.** Illustration of generating and selecting waypoints based on the results of NLU.

We note that the complexity of Algorithm 1 is positively proportional to the number of task zones, the sizes of the candidate waypoints of the task zones, as well as the complexity of the chosen path planning algorithms for the task zones.

---

**Algorithm 1** Waypoints generation and selection based on NLU results

---

Input: structured commands $\{Act_i, Loc_i\}$, $(i = 1, 2, \ldots, n)$;
  start point $S_0$ and end point $E_{n+1}$.
Output: waypoints $\{WP_j\}$ and connections.
1 Obtain the sequence of tasks $\{Task_i\}$, $(i = 1, 2, \ldots, n)$;
2 For $1 \leq i \leq n$
3  Locate the corresponding risky zone $Z_i$;
4  Generate a set of candidate waypoints $\{WP_k(Z_i)\}$;
5  Select a path planning algorithm $Alg_i$;
6  Select the start point $S_i \in \{WP_k(Z_i)\}$ closest to $E_{i-1}$ or $S_0$;
7  Select the end point $E_i \in \{WP_k(Z_i)\}$ closest to $O_{i+1}$ or $E_{n+1}$;
8 End
9 Connect $\{S_0, S_1, E_1, \ldots, S_n, E_n, E_{n+1}\}$.

---

## 4. Simulations and Results

### 4.1. Environmental Settings

In this paper, we designed a simulation environment to verify the proposed method using the XTDrone platform based on the Robot Operating System (ROS), PX4 and Gazebo. A default quadrotor UAV model was loaded into the simulation, and then the UAV was controlled using the off-board mode. The workflow of the simulation is shown in Figure 6.

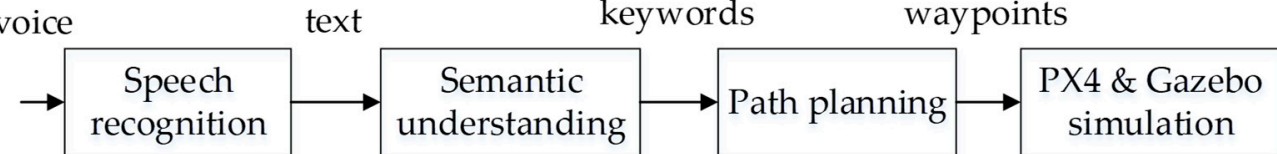

**Figure 6.** Workflow of the simulation.

First of all, we used a Linux SDK provided by the IFLYTEK open platform for speech recognition [42]. Then, a text message was obtained and sent to the semantic understanding module for keywords extraction. As discussed in Section 3.3, the recognized text of the voice command was annotated into a sequence of "action + location". Then, the semantics of the command could be correlated with the entities in the given map. Accordingly, the path planning module orchestrated the segmented paths generated by RTT or Dubins curves. Finally, the waypoints were sent to Gazebo and PX4 for simulation and visualization. The simulation was run on a desktop computer equipped with a CPU of Intel(R) Core(TM) i7-9750H 2.60GHz and a GPU of NVIDIA GeForce RTX 2070.

### 4.2. Simulation Results

We carried out two simulations to test the proposed method. The first simulation was a simple obstacle avoidance task, and the second simulation was a more complex reconnaissance and surveillance task.

### 4.2.1. Simulation 1: Obstacle Avoidance

The task environment with four modeled entities is shown in Figure 7. As introduced in Section 3.2, cuboids and cylinders were used to represent obstacles, i.e., all the four entities (1, 2, 3, 4) were obstacles that should be avoided.

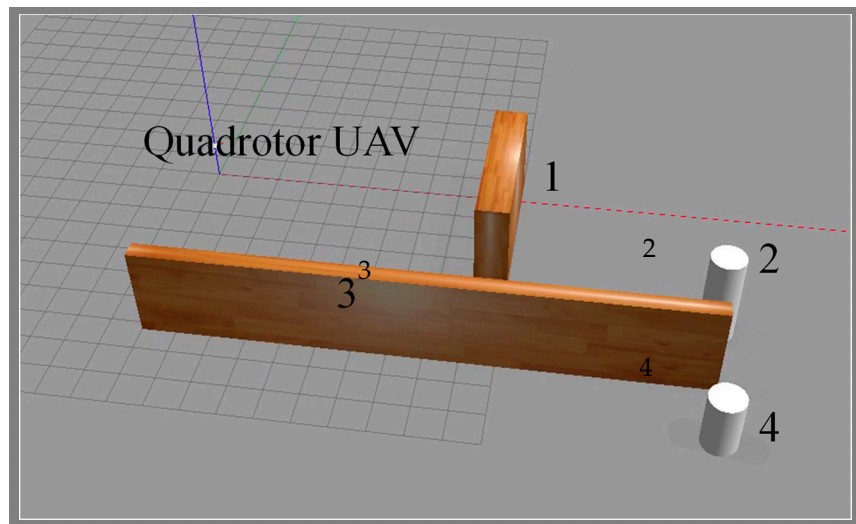

**Figure 7.** An example of task environment with four obstacles.

In simulation 1, the uttered command was "avoid obstacle 1, obstacle 2, obstacle 3, obstacle 4". Based on the RNN-based tagging method, the command was tagged as "avoid (VE) obstacle (CL) 1(NO), obstacle (CL) 2(NO), obstacle (CL) 3(NO), obstacle (CL) 4(NO)". The semantic combination result was "(avoid/Action) (obstacle 1/Location), (none/Action) (obstacle 2/Location), (none/Action) (obstacle 3/Location), (none/Action) (obstacle 4/Location)". Then, the result was checked, and the final annotated sequence was "(avoid/Action) (obstacle 1/Location), (avoid/Action) (obstacle 2/Location), (avoid/Action) (obstacle 3/Location), (avoid/Action) (obstacle 4/Location)".

In the given map, Algorithm 1 was used to plan the waypoints as shown in Figure 8. The whole path was segmented into several parts. For example, the path for avoiding obstacle 1 was {(7,8), (7,3), (12,3)}, and the path for avoiding obstacle 4 was {(19,−2), (19,−4), (16,−7), (13,−7)}. We note that the UAV was allowed to keep close to the obstacles while flying around them, and both RRT and Dubins curves were used for path planning. The actual flying 3D and 2D trajectories are shown in Figure 9.

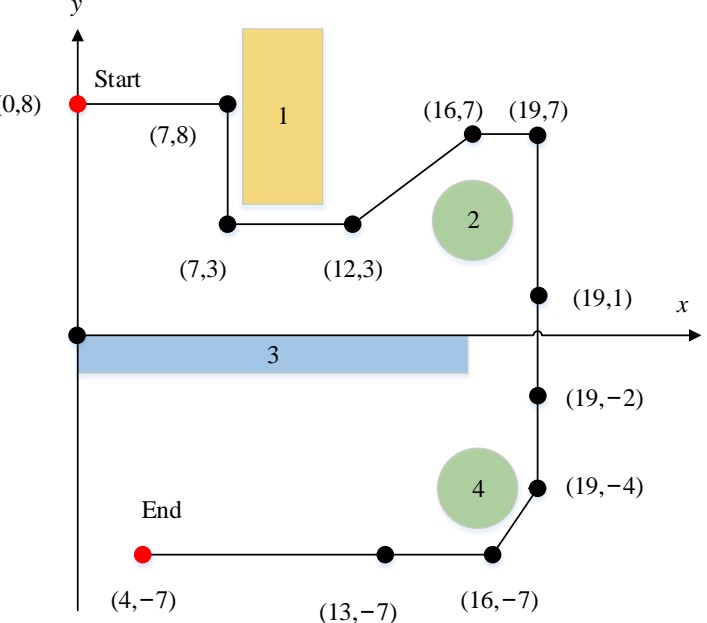

**Figure 8.** Path-planning result of simulation 1.

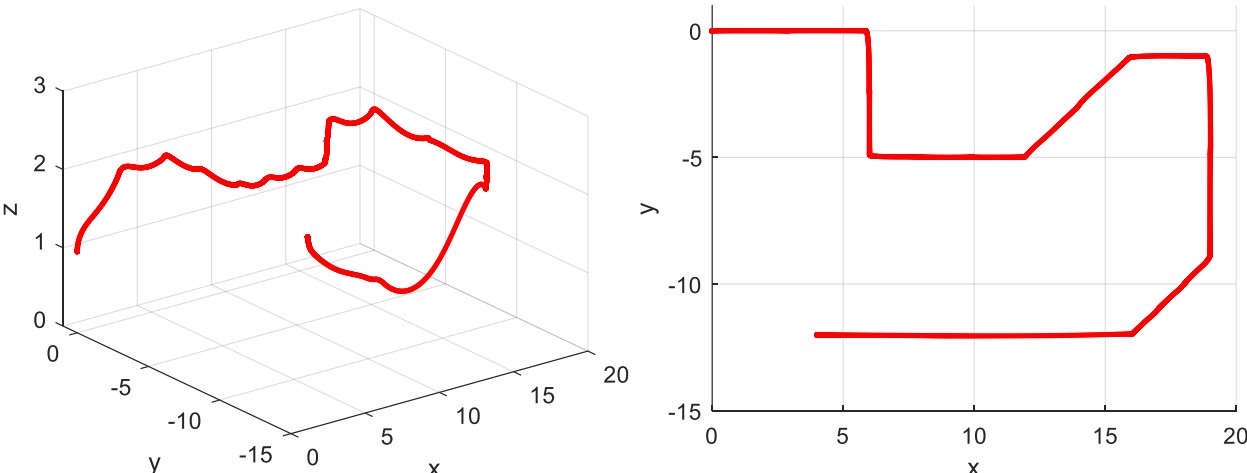

**Figure 9.** UAV Trajectories in the 3D space (**left**) and the 2D projection plane (**right**).

4.2.2. Simulation 2: Reconnaissance and Surveillance

The task environment with four modeled entities is shown in Figure 10. As introduced in Section 3.2, a cuboid was used for obstacle 1, a hemisphere was used for radar 2, a rectangle was used for area 3, and a circle was used for target 4.

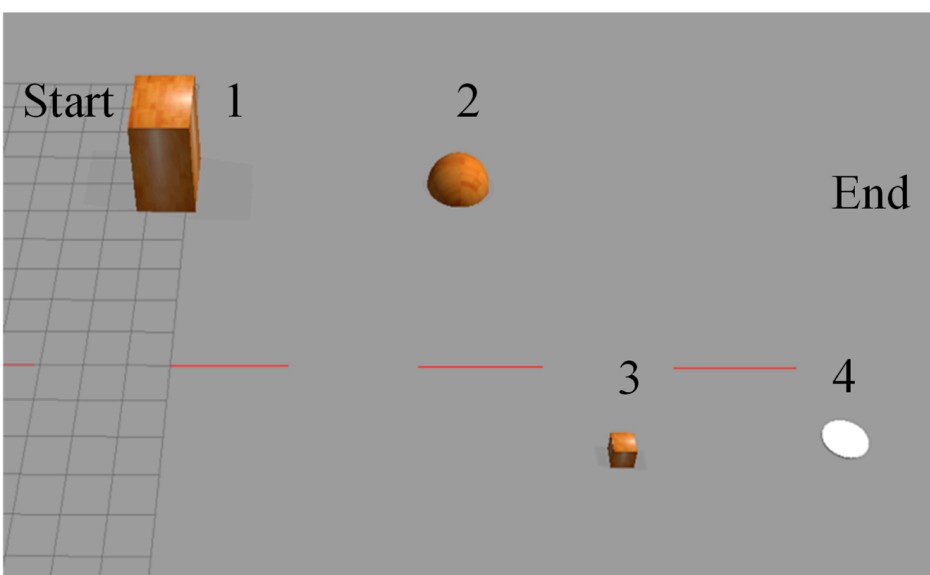

**Figure 10.** An example of task environment with four types of risky task zones (entity 1 denotes an obstacle, entity 2 denotes a radar, entity 3 denotes a reconnaissance area, and entity denotes a surveillance area).

The uttered command was "avoid obstacle 1, avoid radar 2, thorough search area 3, hover track target 4". Based on the RNN-based tagging method, the command was tagged as "avoid (VE) obstacle (CL) 1 (NO), avoid (VE) radar (CL) 2 (NO), thorough (AD) search (VE) area (CL) 3 (NO), hover (AD) track (VE) target (CL) 4 (NO)". The semantic combination result was "(avoid/Action) (obstacle 1/Location), (avoid/Action) (radar 2/Location), (thorough search/Action) (area 3/Location), (hover track/Action) (target 4/Location)".

In the given map, Algorithm 1 was used to plan the waypoints as shown in Figure 11. The actual flying 3D and 2D trajectories are shown in Figure 12. We note that Dubins curves were used for area 2, 3, and 4, while RRT was used for area 1 and other connections between the waypoints.

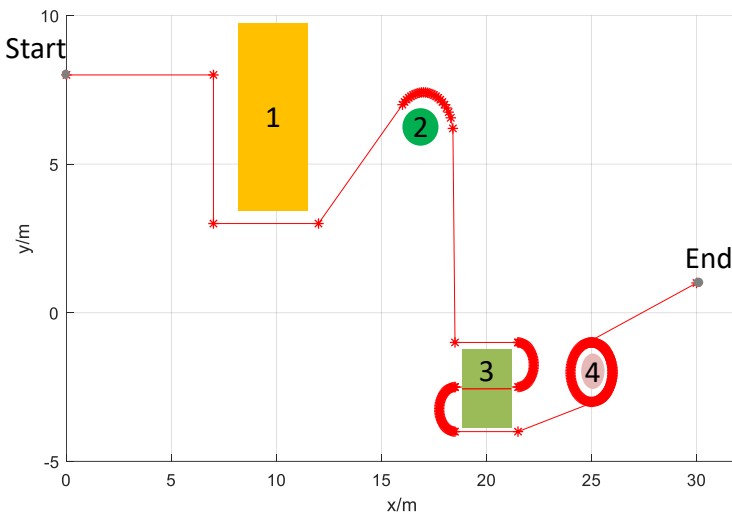

**Figure 11.** Path planning result of simulation 2.

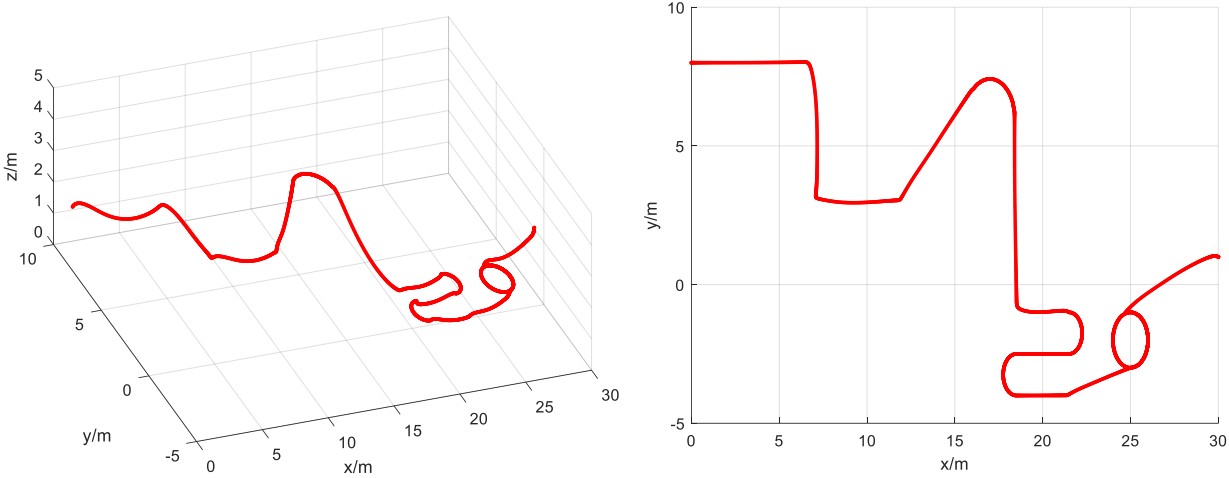

**Figure 12.** UAV Trajectories in the 3D space (**left**) and the 2D projection plane (**right**).

### 4.2.3. Operational Time

We compared the operational time of the proposed method with manual selection of the waypoints. We assumed that the operational time period of the manual approach started from mouse clicking on the given map for waypoints selection, and it ended when a path was generated. The path generation method was simplified as connecting the waypoints with straight lines, which obviously cost much less time than any path planning algorithms. In contrast, the operational time period of our approach started from a human uttering the command, followed by speech recognition and understanding, and it also ended when a path was generated using Algorithm 1. One author of this paper ran the two simulations using the two methods ten times, and the averaged results are given in Table 2.

**Table 2.** Comparison of operational time.

|  | Method | Average Time (In Seconds) |
|---|---|---|
| Simulation 1 | Manual waypoints selection | 9.39 |
|  | Ours | 4.62 |
| Simulation 2 | Manual waypoints selection | 17.66 |
|  | Ours | 6.91 |

As can be seen, the manual approach took a few more seconds than ours. The time was almost spent on reading the map, checking the task zones, and clicking the mouse. This process involved cognitive workload for the human operator, and the workload would increase with task complexity. In contrast, much less cognitive workload was needed for our approach. Most time was spent on uttering the command, i.e., about 4 s for simulation 1 and 6 s for simulation 2. In other words, the cost time increased with the length of the uttered command. The average time of speech recognition along with speech understanding was 0.26 *s*, while the average time of path planning was 0.43 *s*. We note that the comparison was not rigorous due to the biased settings and insufficient subject experiments. However, our method demonstrated time efficiency for solving path planning problems in multi-task scenarios.

## 5. Conclusions

In this paper, we have proposed a novel approach of automatic UAV path planning in risky environments through natural language understanding (NLU). Using RNN-based tagging and semantic combination, a compound command can be segmented into a sequence of paired action and location, and then the keywords of task types and risky zones can be extracted and correlated with the entities of the given map. We have proposed an algorithm to generate candidate waypoints for each task, as well as to select the waypoints based on a distance criterion. We have demonstrated the effectiveness of the proposed method in two simulations. In future work, we will consider a more complex scenario of multi-UAV collaboration, and we will consider the kinematics of fixed-wing UAVs which are more complicated than that of the quadrotor UAVs.

**Author Contributions:** Conceptualization, L.W.; methodology, C.W.; software, Z.Z.; validation, L.W.; formal analysis, C.W.; investigation, C.W.; resources, X.X.; data curation, Z.Z.; writing—original draft preparation, Z.Z.; writing—review and editing, C.W. and Y.Z.; visualization, Z.Z.; supervision, C.W.; project administration, D.Y.; funding acquisition, J.L. All authors have read and agreed to the published version of the manuscript.

**Funding:** This work was supported in part by the Science and Technology Innovation 2030-Key Project of "New Generation Artificial Intelligence" under Grant 2020AAA0108200 and in part by the National Natural Science Foundation of China under Grant 61906203 and 6200612.

**Institutional Review Board Statement:** Not applicable.

**Informed Consent Statement:** Not applicable.

**Data Availability Statement:** Not applicable.

**Conflicts of Interest:** The authors declare no conflict of interest.

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
