# Peer review of "UAV Path Planning in Multi-Task Environments with Risks through Natural Language Understanding"

_drones, doi:10.3390/drones7030147_

Round 1

Reviewer 1 Report

The paper proposed an automatic path planning method through natural language. The combination of natural language processing and UAVs is not satisfactory, giving the reviewer a sense of patchwork concepts. The following comments are given to the authors:

1.The title says risky environment, but the simulation in section 4 does not feel risky enough. It is suggested to improve the expression of the title or make the simulation environment more compact and risky.

2.It is recommended to add some other novel methods of applying deep neural networks to drones in the introduction section, such as 10.1109/TIE.2022.3172764, 10.1109/IROS51168.2021.9636053, 10.15607/RSS.2020.XVI.040 and so on. Some other state-of-the-art UAV planning algorithms can also be added, such as 10.1109/TRO.2022.3160022 or 10.1109/ACCESS.2022.3154037.

3.The advantage of natural language understanding is that it can understand complex and redundant languages, and the language commands designed in this article are too stylized. If it is only this level of order, there is no need to use RNN at all. If natural language understanding is used, at least the UAV can understand different expressions of the same meaning, rather than simple command mapping.

4.There are no physical experiments in this article. It is recommended to change all the 'experiments' in the article to 'simulations' to avoid misunderstandings.

5.The expression of the contributions is not satisfactory. Don't simply express the content of the proposed method, it is recommended to emphasize originality.

6.In section 4, the simulation environment should be described in more detail, such as what performance equipment it is running on, and what is the wheelbase of the UAV and other parameters.

7.The 37th reference only writes a URL link, please double check the format for references.

Reviewer 2 Report

First of all, thank you for submitting the paper to MDPI Drones. After reading the paper, I have some concerns that need to be addressed in detail in the response letter in the next round. 

1. What is the motivation behind the path planning of the UAV using NLP? As you can see, a lot of work has been done for path planning, and the methods proposed by them are quite efficient. How is your work outperforming theirs?

2. Please make a review table. Based on this review table, claim your contribution to your work and mention how efficient your proposed approach is compared to others.

3. Path planning is the same as trajectory optimization. Why haven't you considered the energy consumption of the UAV?

4. The results provided by the authors need to be more comprehensive to fulfil the paper's requirements. Please provide results with practical KPIs, e.g. energy consumption, time, etc. 

5. System Model figure is the missing essential part of the paper. 

6. What is the performance complexity of your algorithm/

7. What is the performance of the system in the worst-case scenario?

8. Last but not least, there are a lot of typos in the paper. Please address them carefully before the next submission. 

I wish you the best of luck with this. 

Reviewer 3 Report

Dear Authors,

the article entitled: UAV Path Planning in Risky Environments through Natural Language Understanding presents an automatic path planning method through natural language that instructs the Unmanned Aerial Vehicle (UAV) with compound commands about the tasks and the corresponding risky regions in a given map. It is very important because the path planning using handcrafted waypoints is inefficient for a multi-task UAV operating in risky environments with bad weather, obstacles, or forbidden zones.

All chapters (abstract, problem statement, method, experiments and results, as well as conclusions) without introduction are well described and they do not raise any doubts. In terms of the literature review is sufficient (37 positions), all of which are papers from recognised scientific conferences and journals, such as: Applied Sciences, Drones, Journal of Intelligent & Robotic Systems, and others. However, I would like to point out that the papers cited are related to the subject of this article (UAV, natural language understanding, path planning, Rapidly-exploring Random Trees (RRT) and Dubins curve). Nevertheless, in the publication make the following change:

• Please describe in general terms the results of the research conducted by authors quoted in the first paragraph of the introduction.

• I propose to extend the literature in the introduction, related to the examples of UAV path plannnig methods such as, for example:

1. Lewicka, O.; Specht, M.; Specht, C. Assessment of the Steering Precision of a UAV along the Flight Profiles Using a GNSS RTK Receiver. Remote Sens. 2022, 14, 6127.

2. Nemra, A.; Aouf, N. Robust INS/GPS Sensor Fusion for UAV Localization Using SDRE Nonlinear Filtering. IEEE Sens. J. 2010, 10, 789–798.

3. Paraforos, D.S.; Sharipov, G.M.; Heiß, A.; Griepentrog, H.W. Position Accuracy Assessment of a UAV-mounted Sequoia+ Multispectral Camera Using a Robotic Total Station. Agriculture 2022, 12, 885.

• The last sentence in the conclusions is not finished.

• Please write sentences impersonally.

To sum up, after taking into account the above amendments (minor revision), I suppose that this article is suitable for publication in Drones.

Round 2

Reviewer 1 Report

The authors have made many improvements compared to previous manuscript. However, there are still some question that the authors need to clarify and improve. 

1. It is recommended to explain the training and simulation under what computing power, such as the CPU and GPU of the computing platform.

2.Figure 8 and Figure 9 are not vector figures and look blurry. It is recommended to change Figure 8 to a picture of the style shown in Figure 11.

3.This path planning algorithm does not take into account the dynamic parameters of the UAV, whether it is possible that the obtained path cannot be followed.

Reviewer 2 Report

Thanks for the revised version, 
Paper is accepted in current form.

Author Response

Thanks for your comments. The manuscript has been further polished by the authors.